# DYNAMICAL SYSTEM EMBEDDING FOR EFFICIENT INTRINSICALLY MOTIVATED AGENTS

## ABSTRACT

Mutual Information between agent Actions and environment States (MIAS) quantifies the influence of agent on its environment. Recently, it was found that the maximization of MIAS can be used as an intrinsic motivation for artificial agents. In literature, the term empowerment is used to represent the maximum of MIAS at a certain state. While empowerment has been shown to solve a broad range of reinforcement learning problems, its calculation in arbitrary dynamics is a challenging problem because it relies on the estimation of mutual information. Existing approaches, which rely on sampling, are limited to low dimensional spaces, because high-confidence distribution-free lower bounds for mutual information require exponential number of samples. In this work, we develop a novel approach for the estimation of empowerment in unknown dynamics from visual observation only, without the need to sample for MIAS. The core idea is to represent the relation between action sequences and future states using a stochastic dynamic model in latent space with a specific form. This allows us to efficiently compute empowerment with the "Water-Filling" algorithm from information theory. We construct this embedding with deep neural networks trained on a sophisticated objective function. Our experimental results show that the designed embedding preserves information-theoretic properties of the original dynamics.

## 1 INTRODUCTION

Deep Reinforcement Learning (Deep RL) provides a solid framework for learning an optimal policy given a reward function, which is provided to the agent by an external expert with specific domain knowledge. This dependency on the expert domain knowledge may restrict the applicability of Deep RL techniques because, in many domains, it is hard to define an ultimate reward function.

On the other hand, intrinsically-motivated artificial agents do not require external domain knowledge but rather get a reward from interacting with the environment. This motivates the study of intrinsically-motivated AI in general, and to develop efficient intrinsically-motivated methods in particular as an alternative and/or complementary approach to the standard reinforcement learning setting.

In recent works, different intrinsic motivation and unsupervised approaches were introduced (Klyubin et al., 2005; Klyubin et al., 2005; Wissner-Gross & Freer, 2013; Salge et al., 2013b; Pathak et al., 2017; Warde-Farley et al., 2018). Among others, the empowerment method reviewed in Salge et al. (2014) uses the diversity of future states distinguishably-achievable by agent as an intrinsic reward. By definition, empowerment of the agent at a given state is the channel capacity between the agent's choice of action and its resulting next state. Previously, this information-theoretic approach has proved to solve a broad range of the AI tasks (Salge et al., 2014; 2013b; Tiomkin et al., 2017; Klyubin et al., 2005). Despite the empirical success, the computational burden of the estimation of empowerment is significant, even for known dynamics and fully observable state spaces, as estimating mutual information between high-dimensional random variables is known to be a hard problem (McAllester & Statos, 2018). Concretely, high-confidence distribution-free lower bounds for mutual information require exponential number of samples. The difficulty in computation has significantly limited the applicability of empowerment in real-life scenarios, ubiquitous in biology and engineering, where an agent observes the environment through sensors (e.g. vision) and does not know an exact dynamical model of the environment.

In this work, we present a novel approach for efficient estimation of empowerment from visual observations (images) in unknown dynamic environments by learning embedding for the dynamic system. The new approach allows us to avoid computationally expensive sampling in high dimensional state/action spaces required for the estimation of MIAS. Efficient computation of empowerment from images opens new directions in the study of intrinsic motivation in real-life environments. As a generic scheme, we structure the interaction between the agent and the environment in a perception-action cycle, shown in Figure 1.

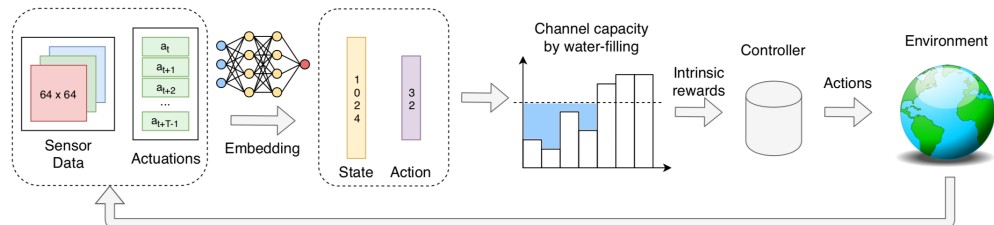

Figure 1: **I**ntrinsically **M**otivated **P**erception and **A**ction **C**ycle. Left-to-right: (a) the observations with the corresponding action sequences are embedded to the latent space by DNNs, where the final states are constrained to be linear in the action sequences (non-linear in the initial states). (b) Gaussian channel capacity between the embedded actions and the embedded final states is computed efficiently by the 'Water-filling' algorithm. (c) an intrinsic reward is derived, triggering the agent to improve its policy for better reward. (d) when the agent operates under the new policy, additional observations from the environment are perceived by the sensors, closing the loop.

While our proposed scheme is broadly applicable, we demonstrate its usage in two different domains: intrinsically motivated stabilization of non-linear dynamical systems and intrinsically-safe deep reinforcement learning. First, the experiments help us verify that the embedding correctly captures the dynamics in the original space. Additionally, our approach for the estimation of MIAS in the embedded space provides a qualitatively similar MIAS landscape compared to the corresponding one in the original space, which allows us to construct meaningful intrinsic reward for reinforcement learning. Our study of the inverted pendulum is especially important because this non-linear model is a prototype for upright stabilization in various biological and engineering systems.

The paper is organized as follows: in Section 2 we compare our work with previous relevant works. Especially, table 1 provides a thorough comparison between existing methods for the estimation of empowerment, and differentiates our work from the existing state of the art. In Section 3 we provide the necessary formal background to present our approach. In Section 4 we present the method, with the implementation described in Section 5. In Section 6 we demonstrate the proposed method by numerical simulations, and compare to the analytical solution derived in the previous work (Salge et al., 2013a). Finally, in Section 7 we conclude, and propose promising continuations of this work.

## 2 RELATED WORKS

In this section, we compare our approach to the existing approaches for the estimation of empowerment. The criteria for comparison are: **(a)** the knowledge of the dynamical model, **(b)** the ability to take high dimensional inputs, (e.g. images/video) and **(c)** the need for sampling to estimate MIAS. The criteria of our choice are relevant because **(a)** the true dynamic model is often inaccessible in real-life environments. **(b)** our IMPAC assumes generic sensor reading, which can be images. **(c)** sampling of high-dimensional data is expensive and it is desired to have an alternative approach for the estimation of MIAS.

One of the contributions of this work is an efficient representation scheme, achieved by deep neural networks, which does not require sampling for the estimation of mutual information. This is the first work that makes it possible to estimate empowerment from images with unknown dynamics and without sampling.

As shown at Table 1, the proposed method makes the least assumptions, (unknown dynamics, visual input, no sampling for the estimation of MIAS) compared to existing methods, which makes it applicable to a broader class of problems.

Table 1: Comparison between existing methods for the estimation of empowerment.

| Method | Unknown dynamics | Visual input | Sampling |
|---|---|---|---|
| Salge et al. (2013a) | no | no | no |
| Mohamed & Rezende (2015) | yes | yes | yes |
| Gregor et al. (2016) | yes | yes | yes |
| Karl et al. (2017) | yes | no | yes |
| Tiomkin et al. (2017) | no | no | no |
| Our work | yes | yes | no |

## 3 PRELIMINARIES

### 3.1 MARKOV DECISION PROCESS

A Markov decision process (MDP) is a discrete-time control model with parameters $S$: the state space, $A$: the action space, $p(s' \,|\, s, a)$: the transition probability model, $r(s, a) \in \mathbb{R}$: the reward function, $p_0$: the initial state distribution, $H$: the horizon in number of steps and $\gamma$: the reward discount factor.

In reinforcement learning, we try to find the optimal policy $\pi$ that maximizes the expected sum of returns along the trajectory:

$$\max_\theta \mathbb{E}\Big[ \sum_{t=0}^{H-1} r(s_t, a_t) \,|\, \pi_\theta \Big].$$

### 3.2 EMPOWERMENT

In this work, we use the established definition of empowerment as a function of state (Klyubin et al., 2005). The empowerment at state $s_t$ is the channel capacity between the action sequence $a_t^{T-1} \doteq (a_t, a_{t+1}, \ldots, a_{T-1})$ and the final state $s_T$.

$$Emp(s_t) = \max_{\omega(a_t^{k-1} \,|\, s_t)} \mathbf{I}(a_t^{k-1}; s_k \,|\, s_t) = \max_{\omega(a_t^{k-1} \,|\, s_t)} \Big\{ H(s_k \,|\, s_t) - H(s_k \,|\, a_t^{k-1}, s_t) \Big\},$$

where $\mathbf{I}(s_k; a_t^{k-1} \,|\, s_t)$ is the mutual information functional, and $\omega(a_t^{k-1} \,|\, s_t)$ is a stochastic policy, described by a probability distribution of action trajectories conditioned on $s_t$.

## 4 PROPOSED APPROACH

The key idea of our approach is to represent the relation between action sequences and corresponding final states by a linear function in the latent space and to construct a Gaussian linear channel in this space. Importantly, even though the relation between action sequences and future states is linear, the dependency of future states on initial states is non-linear. This representation of a non-linear dynamical system enables us to compute empowerment by an efficient "water-filling" algorithm in the latent space, as explained in Section 4.4.

### 4.1 INTERACTION MODEL

We formulate the interaction between the agent and the environment as an MDP (Section 3.1). Each step taken in the environment can be written as:

$$s_{t+1} = f(s_t, a_t, \eta_t), \tag{1}$$

where $s_t$ is the state at time $t$, $a_t$ is the action at time $t$, and $\eta_t$ is assume to be Gaussian process noise. The agent does not know the dynamical model $f$, but rather it obtains through its visual sensors, an image observation $\nu_t$ of the current state $s_t$. Then, after the agent applies an action, $a_t$,

the system moves to the next state, $s_{t+1}$. The agent has access to all previous observation and action trajectories, $\{\tau_i\}_{i=1}^N$, $\tau_i = \{\nu_t^i, a_t^i\}_{t=1}^T$, where $N$ is the number of trajectories and $T$ is the length of each trajectory.

## 4.2 EMBEDDED SPACES

From the collected data, $\{\tau_i\}_{i=1}^N$, we embed tuples in the form of $(\nu_i, a_i^{i+k-1}, \nu_{i+k})$ where $a_i^{i+k-1} = (a_i, a_{i+1}, \cdots, a_{i+k-1})$. Observations and actions are embedded by deep neural networks to the latent spaces $\mathbb{Z} \in \mathbb{R}^{d_z}$ and $\mathbb{B} \in \mathbb{R}^{d_b}$, respectively. Here $d_z$ is the dimension of the embedded state, and $d_b$ is the dimension of the embedded action sequence. These two spaces are related by a linear equation:

$$z_{t+k} = A(z_t)b_t^{t+k-1}, \tag{2}$$

where $z_t$ is the embedded representation of the image $\nu_t$. $A(z_t)$ is a $d_z$ by $d_b$ state dependent matrix that relates the embedded action sequence, $b_t^{t+k-1}$ to the future embedded state, $z_{t+k}$. This model was suggested previously by Salge et al. (2013a) in the original state space with known dynamics.

Our key contribution of this work is the embedding of a generic unknown dynamic system into a latent space that satisfy this model. The architecture for the embedding and the objective functions used to train the networks are provided in Section 5.

## 4.3 INFORMATION CHANNEL IN EMBEDDED SPACE

To compute mutual information between embedded action sequences, $b_t^{t+k-1}$ and embedded states, $z_{t+k}$, we assume that the noise in the system comes from a Gaussian in latent state space: $\eta \sim \mathcal{N}(\mathbf{0}_{d_z \times d_z}, \mathbf{I}_{d_z \times d_z})$. As a result, we end up with an Gaussian linear channel in embedded spaces:

$$z_{t+k} = A(z_t)b_t^{t+k-1} + \eta \tag{3}$$

This particular formulation of dynamic systems in the latent space allows us to solve for the channel capacity directly and efficiently, as described in below.

## 4.4 CHANNEL CAPACITY IN LATENT SPACE

To compute the information capacity of the channel given in Eq. 4.3, we apply the "Water-Filling" algorithm (Cover & Thomas, 2012), which is a computationally efficient method for the estimation of capacity, $C^*$, of a linear Gaussian channel:

$$C^*(z_t) = \max_{p_i} \frac{1}{2} \sum_{i=1}^n \log(1 + \sigma_i(z_t)p_i), \tag{4}$$

under the constraint on the total power, $\sum_{i=1}^n p_i = P$, where $\{\sigma_i(z_t)\}_{i=1}^n$ are the singular values of the channel matrix, $A(z_t)$, and $P$ is the total power in the latent space.

## 5 ARCHITECTURE

The most crucial component of the IMPAC scheme is the embedding from sensor inputs (in our case, images) and action sequences, into latent state and action tensors while remaining consistent with the real environment dynamics. In this section, we organize our discussion around how a single sample: $\{S_t, a_t, \cdots, a_{t+k-1}, S_{t+k}\}$ fits in the training of our deep neural networks, as illustrated in Figure 2. We will explain each individual components of the block diagram and wrap up by discussing the whole picture.

### 5.1 EMBEDDING OF IMAGE OBSERVATIONS

The left side and right side of the block diagram in Figure 2 embeds raw observations (images).

In this work, we take images scaled down to $64 \times 64$ as observations. The image resolution is chosen as a balance of clarity and performance: visual features of the environments are retained

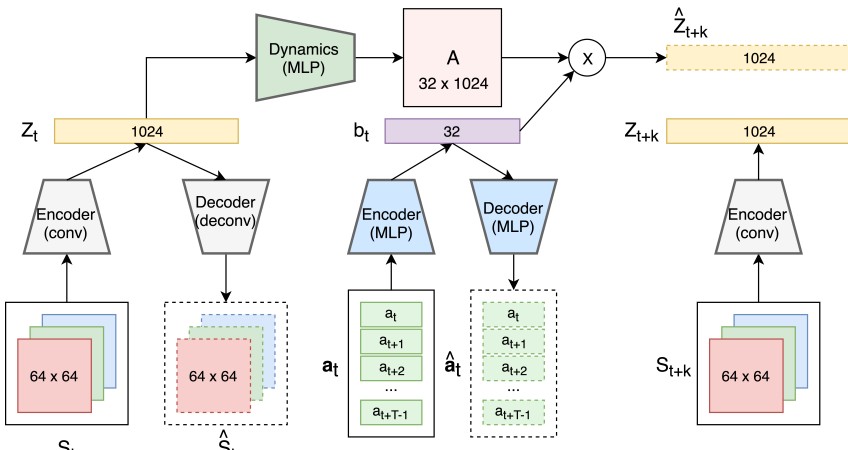

Figure 2: An illustration of data flow around a single training sample.

while training batches can be fit on a single GPU. Multiple images can be concatenated to capture additional information (e.g. velocity). When encoding multi-channel images into latent vectors, we use an auto-encoder and decoder pair parameterized by convolutional neural networks. Given a training sample $\{S_t, a_t, \cdots, a_{t+k-1}, S_{t+k}\}$, both $S_t$ and $S_{t+k}$ are encoded into their respective latent vectors $Z_t$ and $Z_{t+k}$ using a same CNN encoder. To ensure that essential state information is conserved after embedding, we decoded the latent tensor $Z_t$ back to images using a deconvolutional network for correct reconstruction $\hat{S}_t$. The objective is to make the images $S_t$ and $\hat{S}_t$ similar. The encoder and decoder network details are presented in Appendix B.

### 5.2 Embedding of action sequences

The mid-lower section of Figure 2 embeds action sequences.

Standard MDPs (Section 3.1) models environment dynamics in the granularity of single steps, which results in an exponentially large action space over long horizons. Another contribution of this work is that we compute information-theoretic quantities across multiple time-steps without exploding the action space. We consider a $k$-step dynamic model directly by embedding action sequence $\{a_t, a_{t+1} \cdots a_{t+k-1}\}$ into a single action $b_t$ in the latent space (1 step in the latent space corresponds to $k$ steps in the original space). Actions inside the sequence are concatenated into a vector and passed into an auto-encoder to get $b_t$. Again, to ensure that necessary information is retained in the latent action tensor, we decode $b_t$ for reconstruction of $\{a_t, a_{t+1} \cdots a_{t+k-1}\}$. In our implementation, the encoder and decoder are both fully connected MLPs. The detailed information about each layer can be found in Appendix B.

### 5.3 Dynamical model in latent space

The top part of Figure 2 fits a dynamics model linear in action (non-linear in state).

As discussed in Section 4.3, our scheme requires a dynamic model with a particular form. Specifically, given current latent state $Z_t$, the next state $Z_{t+k}$ is linear in the action $b_t$. Mathematically, if latent states have dimension $d_z$ and latent action have dimension $d_b$, we want to find a mapping $A : \mathbb{R}^{d_z} \to \mathbb{R}^{d_z \times d_b}$ such that $Z_{t+k} \approx A(Z_t) \times b$.

The mapping from $Z_t$ to an $d_z \times b_z$ matrix is parameterized by a fully connected MLP with $d_z \times d_b$ output neurons. The final layer is then reshaped into a matrix. (See appendix B)

### 5.4 The entire scheme

In the previous 3 sub-sections, we introduce 3 objectives: 1) Find an embedding of observations that retains their information. 2) Find an embedding of action sequences that retain their information. 3)

Find a dynamics model linear in the latent action. When these three objectives are trained jointly, the neural networks learn towards a latent space that extract information from the original space and learns a dynamics model that is consistent with the actual environment.

$$L = \alpha\mathbb{E}\big[(S_t - \hat{S}_t)^2\big] + \beta\mathbb{E}\big[(a_t - \hat{a}_t)^2\big] + \gamma\mathbb{E}\big[(Z_{t+k} - \hat{Z}_{t+k})^2\big] \tag{5}$$

In practice, regularization on the latent tensors is enforced at training time to encourage a structured latent space. The details about our choice of regularization are discussed in appendix A.

## 6 EXPERIMENTS

### 6.1 INVERTED PENDULUM

Inverted pendulum is a prototypical system for upright bi-pedal walking, and it often serves to check new AI algorithms. Moreover, inverted pendulum is a baseline for the estimation of empowerment in dynamical systems because its landscape is well-studied in numerous previous studies (Salge et al., 2013b; 2014; 2013a; Karl et al., 2017; Mohamed & Rezende, 2015). So, we chose inverted pendulum from the Openai Gym environment for testing our new method, and comparing it to the previous methods (Brockman et al., 2016).

#### 6.1.1 CORRECTNESS OF LATENT DYNAMICS

Prior to applying the proposed model, given by Eq. 4.3, to the estimation of MIAS in the latent space, we verify the correctness of our embedded dynamic model. We compare reconstruction of latent prediction with ground truth observation. In inverted pendulum, two consecutive images are used to capture the state information (angle and angular velocity). When we supply the initial two images for time $t$, along with choices of different action sequences of length $k$, we expect the reconstructions to match the corresponding observations at time $t + k$. As shown in Figure 3 the model is capable of reconstructing the future images from the current images and action sequences.

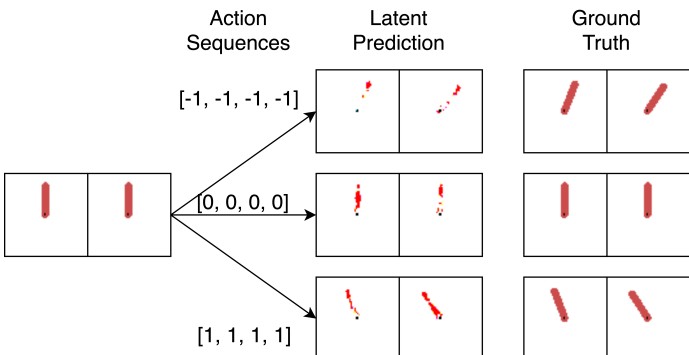

Figure 3: Reconstruction through the latent dynamics in comparison with the ground truth. Starting from the same upright position, a different action sequence is taken in each row. In all 3 cases, reconstruction matches the actual observation after the actions are taken.

#### 6.1.2 EMPOWERMENT LANDSCAPE AND PENDULUM SWING-UP

In this section, we compute the empowerment values and use those as intrinsic reward to train an optimal trajectory for swinging up the pendulum from the bottom position to the upright position. The optimal trajectory is computed by the standard PPO algorithm with reward signal coming solely from empowerment. (Schulman et al., 2017b). Figure 4 compares the empowerment landscape computed under our approach with a previous result:

As shown in Figure 4, at the beginning, the pendulum swings with a small amplitude (the dense black dots around $\theta = \pi\,\text{rad}$, $\dot{\theta} = 0\,\text{rad}\,\text{s}^{-1}$). Once it accumulated enough energy, the pendulum starts traversing the state space, and arrives at the top position, where it stabilises, (dense black points around $\theta = 0\,\text{rad}$, $\dot{\theta} = 0\,\text{rad}\,\text{s}^{-1}$).

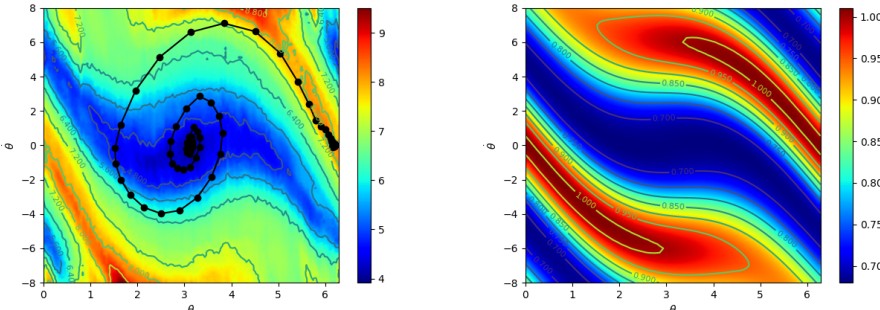

Figure 4: **Left:** Optimal trajectory super-imposed on our empowerment landscape. The pendulum swings up from bottom $(\pi, 0)$ to balance at the top $(0, 0)$.**Right:** Empowerment values calculated with knowledge of the ground truth model (Salge et al., 2013a). This figure shows that the empowerment landscapes are qualitatively similar, which verifies the validity of our approach.

Results from the pendulum experiment effectively demonstrate the validity of our proposed approach. Not only was our method able to create an empowerment value plot similar to that of a previous analytical method, but also these empowerment values train to balance the pendulum at upright position. The pendulum experiment shows that our latent representation effectively captures the information-theoretic properties of the original environment.

## 6.2 SAFETY OF RL AGENT

Another useful application of empowerment is its implication of safety of the artificial agent. A state is intrinsically safe for an agent when the agent has a high diversity of future states, achievable by its actions. This is because in such states, the agent can take effective actions to prevent undesirable futures. In this context, the higher its empowerment value, the safer the agent is. In this experiment, we first check that our calculation of empowerment matches the specific design of the environment. Additionally, we show that empowerment augmented reward function can affect the agent's preference between a shorter but more dangerous path and a longer but safer one.

**Environment:** a double tunnel environment implemented with to the OpenAI Gym API (Brockman et al., 2016). Agent (blue) is modeled as a ball of radius 1 inside a 20×20 box. The box is separated by a thick wall (gray) into top and bottom section. Two tunnels connect the top and bottom of the box. The tunnel in middle is narrower but closer to the goal compared to the one on the right.

**Control:** In each time step, the agent can move at most 0.5 unit length in each of x and y direction. If an action takes the agent into the walls, it will be shifted back out to the nearest surface.

**Reset criterion:** each episode has a maximum length of 200 steps. The environment resets when time runs out or when the agent reaches the goal.

Since the tunnel in the middle is narrower, the agent is relatively less safe there. The effectiveness of the control of the agent is damped in 2 ways:

1. In a particular time step, it's more likely for the agent to bump into the walls. When this happens, the agent is unable to proceed as far as desired.

2. The tunnel is 10 units in length. When the agent is around the middle, it will still be inside the tunnel in the 5-step horizon. Thus, the agent has fewer possible future states.

We trained the agent with PPO algorithm (Schulman et al., 2017a) from OpenAI baselines (Dhariwal et al., 2017) using an empowerment augmented reward function. After a parameter search, we used discount factor $\gamma = 0.95$ over a total of $10^6$ steps. The reward function that we choose is:

$$R(s, a) = \mathbf{1}_{goal} + \beta \times Emp(s)$$

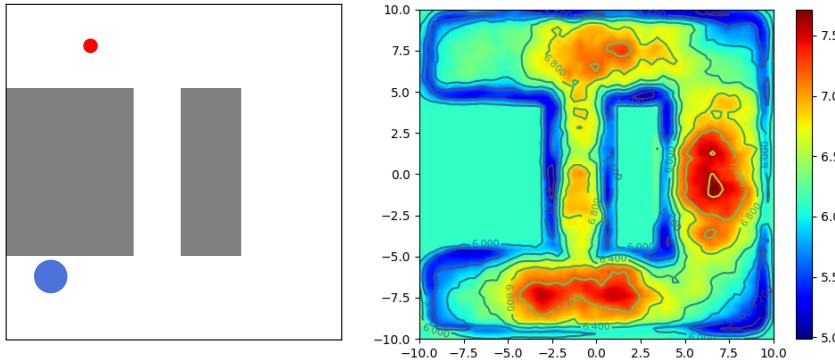

Figure 5: **Left sub-figure**: Double tunnel environment. The goal is marked in red. The agent in blue. **Right sub-figure**: Empowerment landscape for the tunnel environment. The values of empowerment reduce at the corner and inside the tunnel where the control of the agent is less effective compared to more open locations.

where $\beta$ balances the relative weight between the goal conditioned reward and the intrinsic safety reward. With high $\beta$, we expect the agent to learn a more conservative policy.

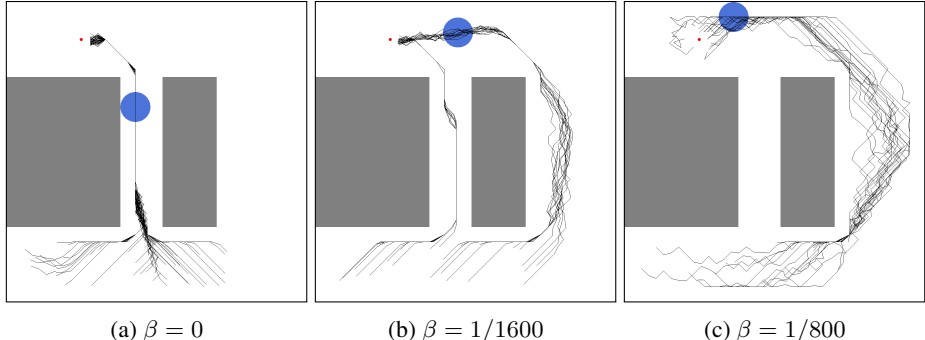

(a) $\beta = 0$        (b) $\beta = 1/1600$        (c) $\beta = 1/800$

Figure 6: Trajectories of trained policy. As $\beta$ increase, the agent develops a stronger preference over a safer route, sacrificing hitting time.

The results from the tunnel environment again support our proposed scheme. First, the empowerment landscape matches our design of the environment. Second, high quality empowerment reward successfully alters the behavior of the agent.

## 7 DISCUSSION

Intrinsically motivated artificial agents do not rely on external reward and thus, do not need domain knowledge for solving a broad class of AI problem such as stabilization, tracking, etc. In this work, we introduce a new method for efficient estimation of a certain type of information-theoretic intrinsic motivation, known as empowerment. The learnt embedding representation reliably captures not only the dynamic system but also its underlying information-theoretic properties.

With this work, we also introduced the IMPAC framework, which allows for a systematic study of intrinsically motivated agents in broader scenarios, including real-world physical environments like robotics. Future works on physical environment will better demonstrate the usefulness and general applicability of empowerment as an intrinsic motivation.

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

## A    DETAILS OF TRAINING ALGORITHM

1. Collect trajectories using random policy and random reset.
2. Separate out tuples $(S_t, a_t, a_{t+1}, \cdots, a_{t+k-1}, S_{t+k})$ from trajectories. For simplicity of notation, let $a_t^k$ represent the concatenated action sequences $a_t, a_{t+1}, \cdots, a_{t+k-1}$
3. Use randomly initialized encoder networks to map the observations and action sequence into latent space. This gives us tuples $(Z_t, b_t^k, Z_{t+k})$.
4. Use randomly initialized MLP network to get the corresponding transformation matrices $A(Z_t)$.
5. Calculate the predicted next state in latent space $\hat{Z}_{t+k} = A(Z_t)b_t^k$
6. Use randomly initialize decoder networks to reconstruct images and action sequences from latent vectors.
   $\tilde{S}_t = Dec(Z_t)$
   $\tilde{a}_t^k = Dec(b_t^k)$
   $\tilde{S}_{t+k} = Dec(\hat{Z}_{t+k})$ Note that $\tilde{S}_{t+k}$ is reconstructed from latent prediction.
7. Calculate the following loss terms:
   (a) Observation reconstruction error: $L_{obs} = ||S_t - \tilde{S}_t||_2^2$
   (b) Action sequence reconstruction error: $L_{action} = ||a_t^k - \tilde{a}_t^k||_2^2$
   (c) Prediction error in latent space: $L_{latent} = ||Z_{t+k} - \hat{Z}_{t+k}||_2^2$
   (d) Prediction error in original space: $L_{org} = ||S_{t+k} - \tilde{S}_{t+k}||_2^2$
8. In additional to the loss terms, we add regularization terms to prevent latent vectors from shrinking. This help us get consistent and comparable empowerment values across different trials and even different environments.
   (a) Regularization of latent state: $Reg_z = |1 - \frac{|Z_t|_2^2}{d_z}|$
   (b) Regularization of latent action: $Reg_b = |1 - \frac{|b_t^T|_2^2}{d_b}|$
9. Finally, we use batched gradient descent on the overall loss function to train all the neural networks in the same loop.

$$L = \alpha_{obs}L_{obs} + \alpha_{action}L_{action} + \alpha_{latent}L_{latent} + \alpha_{org}L_{org} + \alpha_{reg}(Reg_z + Reg_b)$$

For both of our experiments, we chose

$$\alpha_{obs} = \alpha_{org} = 100$$
$$\alpha_{action} = 10$$
$$\alpha_{latent} = \alpha_{reg} = 1$$

and were able to produce desired empowerment plots.

## B    DETAILS ON NEURAL NETWORK LAYERS

### B.1    CONVOLUTIONAL NET FOR IMAGE ENCODING

(h1) 2D convolution: 4 filters, stride 2, 32 channels, ReLU activation
(h2) 2D convolution: 4 filters, stride 2, 64 channels, ReLU activation
(h3) 2D convolution: 4 filters, stride 2, 128 channel, ReLU activations
(h4) 2D convolution: 4 filters, stride 2, 256 channel, ReLU activations
(out) Flatten each sample to a 1D tensor of length 1024

### B.2    DECONVOLUTIONAL NET FOR IMAGE RECONSTRUCTION

(h1) Fully connected layer with 1024 neurons, ReLU activation
(h1') Reshape to $1 \times 1$ images with 1024 channels
(h2) 2D conv-transpose: 5 filters, stride 2, 128 channels, ReLU activation
(h3) 2D convolution: 5 filters, stride 2, 64 channels, ReLU activation
(h4) 2D convolution: 6 filters, stride 2, 32 channel, ReLU activations
(out) 2D convolution: 6 filters, stride 2, $C$ channel

### B.3 MLP FOR ACTION SEQUENCE ENCODING

(h1) Fully connected layer with 512 neurons, ReLU activation
(h2) Fully connected layer with 512 neurons, ReLU activation
(h3) Fully connected layer with 512 neurons, ReLU activation
(out) Fully conected layer with 32 output neurons

### B.4 MLP FOR ACTION SEQUENCE RECONSTRUCTION

(h1) Fully connected layer with 512 neurons, ReLU activation
(h2) Fully connected layer with 512 neurons, ReLU activation
(out) Fully conected layer with $k \times d_a$ neurons, tanh activation then scaled to action space

### B.5 MLP FOR TRANSITION MATRIX A

(h1) Fully connected layer with 1024 neurons, ReLU activation
(h2) Fully connected layer with 4096 neurons, ReLU activation
(h2) Fully connected layer with 8192 neurons, ReLU activation
(out) Fully conected layer with $d_z \times d_b$ ($1024 \times 32$) neurons

