# OpenReview forum: "Dynamical System Embedding for Efficient Intrinsically Motivated Artificial Agents"
_ICLR.cc/2020/Conference — Reject_

### Official Review · AnonReviewer2 · 2019-10-15
**Official Blind Review #2**

**Rating:** 1

**Review:**

This paper proposes a model to embed states and action sequences into latent spaces in order to enable efficient estimation of empowerment in a reinforcement learning system.

The paper shows some interesting experimental results. But overall, it is not ready to publish yet for the following reasons:

1. The technical section is lack of important description/theorems/derivation/etc that are necessary to support the claims.

What is the detailed definition of empowerment, i.e., how to spell out the formula of mutual information I? What is the distribution of action sequences and states? What is the policy function \pi? How is \pi related to or different from \omega? How is the learned empowerment used in (training) policy?

The authors proposed a parametrization in equation-3 for state transition and claimed that this parametrization yields ``an efficient estimation of MIAS as explained in the next section’’, but did not give any explanation throughout the paper. Does it mean the equation-4 is easy to solve? Does equation-4 end up that way because of the parametrization in equation-3? Why so?

What is the water-filling algorithm? How does it associate the capacity with the empowerment? Elaboration is needed for this bit.

The authors claim that putting more weight on empowerment in reward ends up in a more conservative policy, but didn’t give any technical justification. They indeed mention that ``a state is intrinsically safe for an agent when the agent has a high diversity of future states’’ and that ``the higher its empowerment value, the safer the agent is’’. The former is a hypothesis and the latter needs technical/derivation support---e.g., why empowerment is correlated with the diversity? The interesting experimental results in figure-6 seem to support the authors’ claim, but precise technical justification is needed.

2. Some technical description/argument should be more precise and accurate.

The authors claim that they ``observes the current state through its visual sensors’’---but the actual state (i.e. the exact angle and height) can’t be observed and the visual sensor data is only an approximation, so the correct claim should be something like: we observe the visual representation of the actual state.

The authors claim that the ``existence of such representation ... is one of the contributions of our work’’---the existence of something shouldn’t be a contribution of a technical paper, what can be a contribution is the proof of its existence.

The authors claim that they ``inject Gaussian noise … into the latent space.’’ This is very confusing: it sounds like (1) there wasn’t any randomness in this formulation; (2) the estimation of empowerment is difficult because of that and (3) the authors added the Gaussian noise to enable the efficient estimation. However, based on my understanding after reading multiple times, I guess what actually happened is: (1) there should be randomness and the noise can be anyway distributed and (2) the authors assumed it is Gaussian so it is simple enough to yield an efficient estimation. The authors should really clarify this.

3. There are also some typos that may confuse readers.

The authors mentioned that ``a linear Gaussian channel given by Eq. 4.3’’---is it section-4.3 or equation-3?

In appendix, what is d_z and d_b?


**Experience Assessment:**

I do not know much about this area.

**Review Assessment: Checking Correctness Of Derivations And Theory:**

N/A

**Review Assessment: Checking Correctness Of Experiments:**

I carefully checked the experiments.

**Review Assessment: Thoroughness In Paper Reading:**

I read the paper at least twice and used my best judgement in assessing the paper.

---

> ### Author Response · Authors · 2019-11-15
> **Response to Review #2**
>
> We appreciate your feedbacks and they helped in our revision of the paper. We focused on making the terms clearer and more intuitive. Here are some additional clarifications:
>
> #1 Empowerment is defined as the maximal of mutual information between action sequences and resulting final states. In previous studies on empowerment in dynamical systems, cited in the paper, it was shown that empowerment can be used as a utility function for stabilization of non-linear dynamical systems. Stabilization with empowerment assumes two policies: $\omega$, and $\pi$. The former is a probing policy, serving for the estimation empowerment, Eq, (1). The latter is a control policy which can be computed by any reinforcement learning approach, given empowerment values. In this work we show an efficient method for the estimation of empowerment in unknown dynamics. The main contribution of this paper is a novel scheme for embedding raw images to the latent space, where the challenging optimization problem given by Eq. (1), is transformed to a convex optimization problem, given by Eq. (4). This optimization problem allows us to find a solution to Eq. (1) by a super-efficient, line search. This problem is known as Water-Filling, which is given by Eq. (5). In the paper, we provided a reference for the detailed derivation of the solution to this problem, (Cover & Tomas, 2012).
>
> #2 It was shown previously, as cited in the paper, that empowerment is decreased when an agent has less control over its environment. This might happen e.g., in narrow tunnels, and in proximity to obstacles. We utilized this property to address safety, assuming that when an agent lacks control it cannot prevent dangerous situations, and as a result, is more vulnerable. As mentioned in the introduction, empowerment is directly related to the diversity of achievable states, which also is seen by the formal definition of empowerment, Eq (1).
>
> #3 d_z and d_b refer to the dimension of latent spaces. They are first introduced in Section 4.2
>
> Thanks again for pointing out the issues with our paper. As we have revamped our writing significantly, we will appreciate a second thought on the rating.

---

### Official Review · AnonReviewer1 · 2019-10-20
**Official Blind Review #1**

**Rating:** 3

**Review:**

This paper proposes to take advantage of a known result on the channel capacity of a linear-Gaussian channel in order to estimate the empowerment and maximize mutual information between  policies (action sequences) and final states (given  initial states). The idea is to map the raw action sequences and states to a latent space where learning would force that linear property to be appropriate.

I like the general idea of the paper (as stated above) along with its objectives but I have several concerns.

First, I need to be reassured that we are computing the right quantity. Channel capacity is the maximum mutual information (between inputs and outputs) over the input distribution, whereas I had the impression that empowerment would be this mutual information, and that we want to increase it, but not necessarily reach its maximum over all possible policies: it would usually be one of the terms in an objective function (e.g. here we have reconstruction error, and in practice there would be some task to solve in addition to the exploration reward). One way to see this problem in the given formulation is that the C* objective only depends on the matrix A (which encapsulates the conditional density of z_{t+1} given the trajectory) and it does not depend at all on the distribution of the trajectory itself! This is weird since if we are going to use this as reward the objective is to improve the trajectory. What's the catch? So either I misunderstand something (which is quite possible) or there is something seriously wrong here.

I am assuming that the training objective for the encoder is a sum of the reconstruction error and of C*. But note how this does not give a reward for policies, as such. This is a bit strange if the goal is to construct an exploratory reward!

A less radical comment is: have you verified that the linear relationship between z_{t+k} and the b actually holds well? In other words, is the encoder able to map the raw state and actions to a space where the linearity assumption  is correct, and thus where equation (3) is satisfied.

Figure 3 has something weird,  probably one of the two sequences of -1's should be a sequence of +1's.

Figure 4 is difficult to interpret, the caption should do a better job.

The experiment on the safety of the RL agent is weak. I don't see the longer path as safer, here. And the results are not very impressive, since the agent is only doing what it's told, i.e., go in areas with more options (more open areas) but there is no reason to believe that this is advantageous, here.

Finally, what would make this paper much more appealing is if the whole setup led to learning better high-level representations (how to measure that is another question, but it is a standard kind of question in representation learning papers).

Related work:

I don't understand why in the abstract the authors refer to sampling-based methods as requiring exponentially many samples. This is not generally the case for sampling based methods (e.g. think of VAEs). I suppose the authors refer to something in particular but it was not clear to me what.

References: in the intro, you might want to refer to the contrastive methods and variational methods to maximize mutual information between representations of the state and representations of the actions, e.g.,
 Thomas et al, 2018, 1802.09484
 Kim et al, 2018, arXiv:1810.01176
 Warde-Farley et al, 2018, arXiv:1811.11359


Evaluation: for now I suggest a weak reject but I am ready to modify my score if I am convinced that my main concerns were unfounded.


**Experience Assessment:**

I have published one or two papers in this area.

**Review Assessment: Checking Correctness Of Derivations And Theory:**

I carefully checked the derivations and theory.

**Review Assessment: Checking Correctness Of Experiments:**

I carefully checked the experiments.

**Review Assessment: Thoroughness In Paper Reading:**

I read the paper thoroughly.

---

> ### Author Response · Authors · 2019-11-15
> **Response to Review #1**
>
> We appreciate the insightful feedback you provide which can help us communicate the work more effectively. We have posted a revision of our work and additional explanations below.
>
> #1. You expressed a concern about the validity of our overall scheme and optimization objectives.
> To make things clear, we list the key stages in our workflow:
> (1) Estimation of empowerment from trajectories
>   (a) Fitting the representations z, b and latent dynamics such that is linear in b
>   (b) Compute empowerment by 'Water-Filling'
> (2) Use the empowerment values as intrinsic reward for control tasks
>
> Key definition: Empowerment is defined as the maximal mutual information (channel capacity) between action sequences and future states. Given a specific horizon T, this quantity is a function of state z. It does not depend on trajectory or actions. Using it as intrinsic reward, we encourage the agent to operate in a more controllable region. Note that (1-a), (1-b) and (2) each defines a different optimization problem. As shown in Figure 1, these three optimizations are done independently, and they are, by design, separate steps.
>
> #2. You pointed out the lack of verification for the linear model. The revision emphasizes such verifications. First, we verify the model by reconstruction (in Figure 3) The catch here is that, as shown in Figure 2, encoders and decoders are never trained to optimize for reconstruction. This means the prediction of our linear model is accurate, and properly represents the original nonlinear dynamics. Second, we compare our empowerment estimation with previous works on the nonlinear inverted pendulum, as shown in Figure 4. Solving for channel capacity is not a trivial task, and in our case, it relies on a high-quality linear dynamics model. The fact that our method was able to create an empowerment value plot similar to that of previous analytical method justifies the validity of our linear model. Finally, previous work shows that pendulum learns to balance at the top using only empowerment signal. Since we achieve the same behavior using our empowerment estimation (shown by the black trajectory in Figure 4), we know that our entire method is valid.
>
> #3. Thanks for pointing out the typo.
>
> #4. You pointed out that the explanation for Figure 4 is not sufficient. Additional explanation has been added.
>
> #5. We use this experiment as another testbed to verify our empowerment estimation. The relation between empowerment and the safety of the agent is established in previous works: https://doi.org/10.3389/frobt.2017.00025, https://doi.org/10.7551/978-0-262-31709-2-ch018 Our intention was not to reiterate the old findings, but for this particular experiment, safety can be interpreted in this way: When the agent collides with the wall, its control is less effective (low empowerment). Since collision is dangerous, we prefer states with high empowerment and avoids the tunnel, where its empowerment reduces. Our empowerment estimation successfully identifies areas with more options and avoids narrow tunnels. This shows that the underlying representation of the dynamics system is valid.
>
> #6. Our model, A(z), given by Eq. (3), is an interpretable representation of an original nonlinear dynamics. The advantage of our representation is that it allows us to estimate the maximal mutual information between input and output efficiently. In this sense, this submission introduces, for the first time, an optimal representation learning of an arbitrary dynamic from observations only. In contrast to the previous works, the current work estimates mutual information of nonlinear dynamics by convex optimization, rather than by Monte Carlo sampling. We believe this submission perfectly match the core theme of ICLR conference.
>
> #7 We referred to the fact that any high-confidence distribution-free lower bound of mutual information requires an exponential number of samples. In this work we propose a framework how to approximate the maximal mutual information between action sequences and future states overcoming the requirement on an exponential number of samples.
>
> #8 Thanks for pointing out some very relevant sources!
>
> We appreciate your second thought on your rating of our work!

---

### Official Review · AnonReviewer3 · 2019-11-01
**Official Blind Review #3**

**Rating:** 3

**Review:**

This paper proposes an approach based on empowerment for reinforcement learning applicable to the cases that the dynamical system is unknown. The model is estimated by a water filling algorithm and is evaluated on two RL tasks.

Training of RL agents via on empowerment and intrinsic rewards is an important alternative to conventional training algorithms. The paper is tacking an important paper.
The paper is weak in terms of writing and motivation. Empowerment on section 3.2 could have been explained more intuitive and more thoroughly the make the paper self-contained.
Moreover, the paper lacks motivation of the design choices. It seems to be a combination of a few recent techniques in machine learning or statistics that are mechanically attached to each other without sufficient justification or intuition.
The paper keeps claiming to solve AI however what it actually experimented on RL safety and/or one synthetic environment. They are not really "AI benchmark problems". I'd rather the paper focuses on its contribution: direct and concise.
Moreover, the experiments are not sufficient to support the paper or investigate how much each part is contributing to the success.

**Experience Assessment:**

I do not know much about this area.

**Review Assessment: Checking Correctness Of Derivations And Theory:**

I assessed the sensibility of the derivations and theory.

**Review Assessment: Checking Correctness Of Experiments:**

I assessed the sensibility of the experiments.

**Review Assessment: Thoroughness In Paper Reading:**

I read the paper at least twice and used my best judgement in assessing the paper.

---

> ### Author Response · Authors · 2019-11-15
> **Response to Review #3**
>
> We appreciate your feedbacks regarding which we have significantly revamped the original submission.
>
> #1. We added more intuitive explanations in the abstraction and introduction. Hope that they will help make the paper easier to read.
>
> #2. In the introduction and proposed approach section, we added more explanations on how different parts of the design come together. Essentially, our method fills in the hole of existing algorithms for empowerment and tries to combine the strengths of each of them.
>
> #3. We carefully reviewed our overall writing. We’ve made the texts much more precise and direct.
>
> #4. We elaborated on existing experiments to better address their implications.
>
> We will appreciate your second thought on your rating.

---

### Decision · Program_Chairs · 2019-12-19

**Decision:**

Reject

**Comment:**

The paper proposes a novel method for embedding sequences of states and actions into a latent representation that enables efficient estimation of empowerment for an RL system. They use empowerment as intrinsic reward for safe exploration. While the reviewers agree that this paper has promise, they also agree that it is not quite ready for publication in its current state. In particular, the paper is lacking a theoretical justification for the proposed approach, the definition of empowerment used by the authors raised questions, and the manuscript would benefit from more clear and detailed description of the method. For these reasons I recommend rejection.